Distribution-preserving data augmentation

Saran Nurdan Ayse 1 buz@cankaya.edu.tr
http://orcid.org/0000-0002-8652-3392 Saran Murat 1
http://orcid.org/0000-0002-3003-8136 Nar Fatih 2
1 Department of Computer Engineering, Cankaya University , Ankara , Turkey
2 Department of Computer Engineering, Ankara Yildirim Beyazit University , Ankara , Turkey
Ventura Sebastian
Electronic publication date: 2021 May 27
Publication date: 2021
Volume: 7
Electronic Location ID: e571
Received 2021 Feb 10; Accepted 2021 May 10
Copyright: © 2021 Saran et al.
Copyright year: 2021
Copyright holder: Saran et al.
License: This is an open access article distributed under the terms of the Creative Commons Attribution License, which permits unrestricted use, distribution, reproduction and adaptation in any medium and for any purpose provided that it is properly attributed. For attribution, the original author(s), title, publication source (PeerJ Computer Science) and either DOI or URL of the article must be cited.
License URL: https://creativecommons.org/licenses/by/4.0/

Keywords: Machine learning, Deep learning, Data augmentation, Color-based augmentation, Transfer learning

Funding: The authors received no funding for this work.

==============================
In the last decade, deep learning has been applied in a wide range of problems with tremendous success. This success mainly comes from large data availability, increased computational power, and theoretical improvements in the training phase. As the dataset grows, the real world is better represented, making it possible to develop a model that can generalize. However, creating a labeled dataset is expensive, time-consuming, and sometimes not likely in some domains if not challenging. Therefore, researchers proposed data augmentation methods to increase dataset size and variety by creating variations of the existing data. For image data, variations can be obtained by applying color or spatial transformations, only one or a combination. Such color transformations perform some linear or nonlinear operations in the entire image or in the patches to create variations of the original image. The current color-based augmentation methods are usually based on image processing methods that apply color transformations such as equalizing, solarizing, and posterizing. Nevertheless, these color-based data augmentation methods do not guarantee to create plausible variations of the image. This paper proposes a novel distribution-preserving data augmentation method that creates plausible image variations by shifting pixel colors to another point in the image color distribution. We achieved this by defining a regularized density decreasing direction to create paths from the original pixels’ color to the distribution tails. The proposed method provides superior performance compared to existing data augmentation methods which is shown using a transfer learning scenario on the UC Merced Land-use, Intel Image Classification, and Oxford-IIIT Pet datasets for classification and segmentation tasks.

Introduction

Since the first study conducted by Krizhevsky, Sutskever & Hinton (2012) in the ImageNet competition in 2012, deep learning (DL) has been highly successful in image recognition problems. Today convolutional neural networks (CNN) are well-understood tools for image classification as a heavily employed DL approach. CNN’s main strength comes from its ability to extract features automatically from regularly structured data such as speech signals, images, or medical volumes (Georgiou et al., 2020), or even unstructured data such as point clouds (Charles et al., 2017). However, training a DL network with high accuracy and generalization capability requires a large dataset representing the real world. Thus, the performance of deep learning algorithms relies heavily on the variety and the size of the available training data. Unfortunately, it may be challenging to obtain a sufficiently large amount of labeled samples (Wang et al., 2020; Kemker, Salvaggio & Kanan, 2018). In some cases, gathering the data is complicated or even hardly possible. Therefore, training DL becomes challenging due to insufficient training data or uneven class balance within the datasets (Huang & Du, 2005).

One way to deal with an insufficient training data problem is using so-called data augmentation techniques to enlarge the training data by adding artificial variations of it (Simard, Steinkraus & Platt, 2003). Such an enlarged training dataset can be even further extended by adding synthetically generated data (Wong et al., 2019). Data augmentation can be applied directly to the features, or it can be applied to the data source, which will be used to extract the features (Volpi et al., 2018), e.g., CNN can extract features from the enlarged image dataset (Shorten & Khoshgoftaar, 2019). The most challenging work is to improve the generalization ability of the trained model to avoid overfitting. If correctly done, data augmentation techniques can improve the performance and generalization ability of the trained model. Therefore, due to their success, data augmentation techniques are used in many studies that employs machine learning (Ali et al., 2020; Islam, Wijewickrema & O’Leary, 2019; Zheng et al., 2019).

Data augmentation strategies can be divided into three groups as color transformations, geometric transformations, and techniques using neural networks. Methods using color transformations manipulate the pixels’ spectral values by doing operations such as changing the contrast, brightness, color or injecting noise (Takahashi, Matsubara & Uehara, 2020), or applying some filtering techniques (Zhu, He & Zheng, 2020). As a color-based approach, Zhong et al. (2020) proposed a random erasing technique that either randomly puts a filled rectangle or puts a random-sized mask into a random position. Methods using geometric transformations are manipulating pixel positions by doing operations such as scaling, rotation, flipping, or cropping (Shorten & Khoshgoftaar, 2019). In particular, methods based on geometric transformation should be selected according to the target dataset. For example, in CIFAR-10, horizontal flipping is an efficient data enlargement method, but it can corrupt data due to different symmetries in the MNIST dataset (Cubuk et al., 2018). Similarly, as in face recognition samples, if there is a dataset where each face is centered in the frame, geometric transformations give outstanding results (Xia et al., 2017). Otherwise, one should ensure he/she does not alter the label of the image while using these augmentation variants. Moreover, the possibility of distancing the training data from the test data should also be considered (Shorten & Khoshgoftaar, 2019). As with geometric transformations, some color transformations can also distort important color information, changing the image label (Shorten & Khoshgoftaar, 2019). Augmentation methods can also be combined to increase the variety in the resulting augmented images (Cubuk et al., 2018). With a careful setting, these color and geometric transformations help generate a new dataset covering the span of image variations (Howard, 2013). As recent approaches, Mun et al. (2017), and Perez & Wang (2017) proposed to generate synthetic images which retain similar features to the original images samples using various types of Generative Adversarial Networks (GAN). However, Chen et al. (2020) observed that the cost of training is time-consuming while the variability of data produced is often limited. Data augmentation methods can produce good results with different parameters in different types of problems. Even a single augmentation method is employed, the best parameters should be determined. For a combination of augmentation methods, determining which data augmentation methods to use and their execution order in addition to their optimal parameters is challenging. Thereby, in Cubuk et al. (2018), the auto augmentation method was proposed that searches many augmentation algorithms to find the highest validation accuracy automatically.

In Fig. 1A, an image that contains a blue lake and sky and green trees is shown. Plausible variations of this lake image with some red trees and color changes in clouds and lake are presented in Figs. 1B–1C. Note that, there are differences in these two plausible images, i.e., there are more red trees in the Fig. 1C compared to Fig. 1B. Although Fig. 1D is also a plausible image, it contains a limited variation. These plausible images are more probable to occur in the real world than the unplausible images given in Figs. 1E–1F. The color distribution of the lake image is shown as 3D scatter plot and color channel histograms in Fig. 2. As seen in Fig. 2, some colors frequently occur in an image, while some are rare. For example, the blue lake (mode in distribution and its surroundings) is always seen, but a lake that goes into purple (tails of the distribution) is rare. Mainly images at the tail of the color distribution, in particular, are also infrequent, yet they are plausible.

Figure 1 Example plausible and unplausible images.

(A) Original image. (B–C) Plausible images. (D) Gamma corrected image. (E) Histogram equalized image. (F) Random erased image.

Figure 2 Color distribution for the image in Fig. 1A.

(A) Color distribution as 3D scatter plot. (B) Color channel histograms.

The majority of the data augmentation methods have been originated from image processing and computer graphics domains. Methods specifically targeted to data-augmentation are also proposed in the literature such as random erasing (Zhong et al., 2020), Cutout (DeVries & Taylor, 2017), CutMix (DeVries & Taylor, 2017), MixUp (Yun et al., 2019), and AugMix (Hendrycks et al., 2019). Although these methods are easy to implement and fast, they do not have an inherent mechanism to create plausible image variants. These color-based augmentation methods generally disturb the color distribution of the given image, which leads to the creation of unnatural images, as seen in Figs. 1E–1F. Differently, we propose a distribution-preserving data augmentation method that creates plausible variations of the given image as seen in Figs. 1B–1C. Therefore, this study aims to achieve diversity on the enlarged training data by creating augmented images aligned with the image color data distribution. We were inspired by the mean-shift process by Fukunaga & Hostetler (1975) and Comaniciu & Meer (2002) to obtain a distribution-preserving data augmentation mechanism. The mean-shift process seeks the local mode without estimating the global density, hence avoiding a computationally intensive task. Unlike the mean-shift process, we get a path towards the tails in a density decreasing manner in our method. In the original mean-shift, the data gets denser, and the mean-shift path becomes smooth as it goes towards the distribution mode. However, as we go in the opposite direction, the data becomes increasingly sparse, which can cause the obtained path to act chaotically. We developed a regularized density decreasing direction to create paths from colors of the original image pixels to the image data distribution tails to prevent this. Then we shift the pixel colors to any point in the obtained path so that modified pixel colors will be in alignment with the color distribution of the original image. Thus, the proposed data augmentation method considers the color distribution of the image to produce plausible images by changing colors in this way. Since we shift the color in the augmented image, we also increase the training data diversity. Source code of the proposed method is shared (https://github.com/msaran1923/dpda, https://github.com/msaran1923/dpdalinusx) to facilitate reproducibility.

The remainder of the paper is organized as follows. First, ‘Materials & Methods’ briefly explains the employed background materials and the proposed method, namely the Density Preserving Data Augmentation (DPDA) method. Then, ‘Experiments & Results’ presents the proposed DPDA method’s performance with various qualitative and quantitative experiments and comparison studies. Afterward, a discussion with some remarks on possible future studies is given in ‘Discussion & Future Works’. Finally, conclusions are stated in ‘Conclusions’.

Materials & Methods

To obtain a density decreasing path, we used the opposite of the mean-shift direction. As we go in the density decreasing direction, the data becomes increasingly sparse, which can cause the obtained path to act chaotically. We enforce the density decreasing path’s smoothness by implementing a regularization on the reverse mean-shift direction to prevent this. Such regularized density decreasing paths for 3 pixels are shown in Fig. 3 as examples. The colors of these 3 pixels are chosen to be close to the tail of the distribution to demonstrate the behavior of the density decreasing path generation. One can easily see that paths are smooth even if they move to extremely sparse regions of the image color distribution. Density decreasing path may contain varying numbers of points where the distance between the consecutive points can also be different. We want to have the same number of points, L, in each density decreasing path (L = 64). We also want to equalize the distance between consecutive points in the density decreasing path. We construct a refined density decreasing path while satisfying these two objectives using cubic spline interpolation on the density decreasing path we found.

Figure 3 Density decreasing paths for Fig. 1A.

(A)–(C) Paths for 3 different data points.

In Fig. 4, a process diagram of the proposed DPDA method is given. It has three main steps: extraction of features, and creating density decreasing paths for once, and creating augmented images. First, an Input Image is given to the DPDA method. Then Color Features are extracted from the given image. Then Density Decreasing Paths are created using the extracted Color Features. Finally, several Augmented Images are created. During the creation of augmented images, Perlin Noise is created C times for a given input image and fused with Density Decreasing Paths resulting in C augmented images.

Figure 4 DPDA process diagram.

Extraction of features

DPDA method works with color images with three channels (red, green, and blue), namely RGB image. For an input RGB image in size W × H there are n pixels where n = WH. If we flatten the pixels of this RGB image, then our feature matrix X has a size of n × d where d = 3. Although n features are sufficient, we further enriched the feature space using the image pyramid approach (Adelson et al., 1984). During the image pyramid generation, we halved the original image in width and height three times. This creates an image pyramid with four levels where each level contains four times fewer pixels than the higher level in the pyramid. We used Lanczos interpolation over 8 × 8 neighborhood for the down-sampling operation (Turkowski, 1990). Using an image pyramid with four levels increases the number of features in X by 32.8%. In Fig. 5A, there are structural missing regions in feature space. This is due to quantization error since decimal parts of colors are quantized in 8 bits RGB images. However, refined feature space in Fig. 5B is denser, and the effect of image quantization errors is reduced, which demonstrates another benefit of the employed image pyramid approach.

Figure 5 Feature space of the image in Fig. 1A, without and with image pyramid.

(A) Features from only image. (B) Features from image pyramid.

Creation of density decreasing paths

Let x be the color of a pixel in the image as a starting point of a path we aim to find. Then probability density function (PDF) on the color feature space X constructed from the image is given in Eq. (1).

(1) P(x)=1n∑j=1nK(x−xj)

where K(.) is a kernel function and xj are data points in X where we used Epanechnikov kernel.

We define a density decreasing path T={x(0),x(1),…,x(i),…} (Fig. 6) where x(i + 1) = x(i) + s(i) for i ≥ 0. Here, x(0) is the starting point of the path and s(i) is a density decreasing direction at point x(i). Also, x(i) is only defined in color space domain where 0 ≤ x(i)r, x(i)g, x(i)b ≤ 255.

Figure 6 Density decreasing path.

Now, we can define the pdf for the point x(i + 1) as below:

(2) P(x(i+1))=1n∑j=1nK(x(i+1)−xj)=1n∑j=1nK(x(i)+s(i)−xj)

Since the points x(i) and xj are constant, we can rewrite above equations as below:

(3) P(x(i+1))=P(x(i)+s(i))=1n∑j=1nK(s(i)−x^j)wherex^j=xj−x(i)

So, x^j points are centered to x(i) where x(i) is shifted to the origin; thus, s(i) becomes a direction vector. Finally, we define a gradient descent direction as below that will lead to a density decreasing path:

(4) x(i+1)=x(i)−∇J(x(i)+s(i))

where J(x(i) + s(i)) is the cost function to minimize which is defined as below:

(5) J(x(i)+s(i))→P(x(i)+s(i))subjecttos(i)∈ΩΩ={∥s(i)∥≤Slengthand1−⟨s(i)Slength,s^(prior)⟩≤Sangle}withs^(i)=s(i)Slengthands^(prior)=s(i−1)∥s(i−1)∥

In Eq. (5), Slength is the maximum length and Sangle is the maximum angle for the direction vector s(i). Here, second constraint is only defined for i > 0 where s^(prior) is the prior direction in unit length and considered as constant. Owing to Eq. (5), density decreasing direction s(i) is regularized in length and orientation to avoid chaotic shifts in sparse data regions.

The gradient descent method is not practical to minimize cost functions with constraints. In such cases, one can use the projected gradient descent (PGD) method, which can minimize a cost function subject to a constraint where this constraint defines a domain (Boyd & Vandenberghe, 2004). Although PGD works fine for cost functions with a single constraint, we can still use it efficiently since our cost function’s length, and orientation constraints only form a single domain, Ω. Therefore, we use PGD to obtain a gradient descent path on the cost function J as defined in Eq. (6).

(6) mins(i)⁡P(x(i)+s(i))subject tos(i)∈Ωx(i+1)=PΩ(x(i)−∇P(x(i)+s(i)))PΩ(xnew)=argmins(i)∈Ω∥(x(i)+s(i))−xnew∥

After doing some algebraic manipulations one can see that s(i) equals to the opposite of the mean-shift direction m(i) such that s(i) = −m(i). First, we will limit the number of iterations in the PGD to L/2 since we aim to find a density decreasing path with a limited number of points. Next, we will stop the PGD iteration if (a) the norm of mean-shift direction is becoming smaller than a tolerance value (Ctolerance) or (b) next point x(i + 1) exiting from the image color space domain. Also, we set default value for Ctolerance as 10−2d. The final density decreasing path generation method is presented in Algorithm 1.

Algorithm 1 Find Density Decreasing Path using PGD.

1: Inputs: x(0), h, IFLANN, L, Ctolerance	
2: for i = 0 : L/2 do	
3: m(i) ← calculateMeanShiftDirection (x(i), h, IFLANN)	
4: if (||m(i)|| < Ctolerance) then	
5: break ⊲ Converged, exits loop	
6: end if	
7: x(i+1) =𝓟Ω(x(i) − ∇P(x(i)+s(i))) ⊲ Move to new point with PGD	
8: if (x(i+1) is out of domain) then	
9: break ⊲ Converged, exits loop	
10: end if	
11: end for	
12: T ← regularize ({x(0), x(1), . . . , x(i)}) ⊲ regularize to equidistant L points	
13: Return T	

Calculation of the mean-shift direction m(i) at point x is as given as below:

(7) m(i)=∑xj∈N(x)K(xj−x)xj∑xj∈N(x)K(xj−x)−x

where K(.) is the kernel function and xj are k nearest neighbours of x. We used FLANN proposed by Muja & Lowe (2014) to have fast k nearest neighbor search operations for efficiency. In this study, we used 256 as the default value of k. Note that, one need to put value of x(i) into the point x in the Eq. (7) given in the Algorithm 1. However, kernel functions require the selection of the bandwidth parameter h. Since each image’s characteristic is different, we estimate bandwidth parameter h from the image to balance differences between images as an approximation to median pair-wise distances to closest points. First, we find the Euclid distances of each pixel with its 4 neighbors. Then, we use Quick Select (Cormen et al., 2009) algorithm to find the median value of these distances as our bandwidth h.

We used the PGD method, which first does a gradient descent step, then back-projection of gradient descent result to the domain Ω. In Fig. 7, blue vectors are prior directions; green vectors are new directions in the domain, and red vectors are new directions out of the domain. Domain Ω is the region between dotted gray lines, determined by the constraints in each gradient descent step.

Figure 7 Example cases for directions and back-projections to domain.

(A) s(1) ∈ Ω. (B) s(2) ∉ Ω. (C) 𝓟Ω(s(2)) ∈ Ω. (D) s(3) ∉ Ω. (E) 𝓟Ω(s(3)) ∈ Ω.

Note that prior direction and new direction form a plane where its normal is the cross product of these two vectors. Therefore, a rotation matrix can be formed, which aligns this normal vector to the canonical z-axis where the prior direction and new direction vectors transform onto xy-plane. Once prior and new directions are rotated, all the back-projection operations can be done in 2D easily then back-projected direction can be rotated back to the original space. We used the method proposed by Möller & Hughes (1999) to construct a rotation matrix that aligns normal vector to z-axis as given in Eq. (8).

(8) v=f×t,u=v‖v‖c=f⋅t,r=(1−c)/(1−c2)→[c+rvx2rvxvy−vzrvxvz+vyrvxvy+vzc+rvy2rvyvz−vxrvxvz−vyrvyvz+vxc+rvz2]

where f is plane normal calculated by cross product of prior direction and new direction, and t is z-axis.

Creation of augmented images

Each pixel has its corresponding density decreasing path with L colors. 0th color (first path node) has the largest color deviation from the original pixel color towards the tail of image color distribution. (L − 1)th color (last path node) equals to original image pixel color. So, we can take a different node (color) from the corresponding path for each pixel of an augmented image. Here all 0 indices will yield to augmented image with the most perturbation, while L − 1 indices will yield to the original image. For each pixel, we can randomly choose an index number between 0 and L − 1, which will lead to different augmented images that allow the generation of any number of augmented images. However, utterly random selection will result in unnatural results. Thus, we want to sample from path nodes in a random but spatially smooth manner. We modified the Perlin noise generator, which is proposed by Perlin (1985) to obtain a smooth but random index map as seen in Fig. 8. Here, we choose parameter values randomly from a predefined range where parameters are roughness (Nroughness), noise scale (Nscale), and noise center (Ncenter). Finally, modified Perlin noise is generated using Cx,y = 0.5 (tanh(Nscale * (Nx,y − Ncenter)) + 1) where Nx,y = Perlin.generate(xNroughness, yNroughness, 1) is original Perlin noise generation function.

Figure 8 Effects of different Perlin noises (top row) on augmented images (bottom row).

Experiments & Results

We conducted qualitative and quantitative experiments using different datasets and DL networks to evaluate the effectiveness of the proposed DPDA method. Training and testing are carried out on a server running Ubuntu Linux with Intel i9 CPU (3.7 GHz), 128 GB RAM, Nvidia RTX 3070 GPU. Python using the Keras API and TensorFlow DL libraries are utilized for training the models. This section describes the datasets and experiments used to obtain qualitative and quantitative results.

Datasets

We used Pxfuel1 for qualitative experiments, and three different datasets for quantitative experiments, namely the UC Merced Land-use Yang & Newsam (2010), the Intel Image Classification (https://www.kaggle.com/puneet6060/intel-image-classification/), and the Oxford-IIIT Pet datasets (Parkhi et al., 2012).

UC Merced Land-use dataset consists of satellite images of size 256 × 256 and 0.3-m resolution that are open to the public. There are a total of 21 classes and 100 images in each class (see Fig. 9).

Figure 9 Example image classes from UCMerced land-use dataset.

(A) Airplanes. (B) Parking lot. (C) Buildings. (D) River. (E) Forest. (F) Agricultural.

The Intel Image Classification dataset contains about 25,000 images of size 150 × 150 pixels, classified under six categories (buildings, forest, glacier, mountain, sea, and street) of natural scenery worldwide (see Fig. 10). The training set is around 17,000 images and the test set is the rest.

Figure 10 Example image classes from Intel Classification dataset.

(A) Forest. (B) Glacier. (C) Sea. (D) Mountain. (E) Street. (F) Buildings.

The Oxford-IIIT Pet dataset includes 37 dog and cat classes with 25 dog and 12 cat categories. There are approximately 200 images for each class with significant variations in image size, exposure, and lighting. The total number of images in the dataset is just over 7,000. The dataset also includes labels as bounding boxes and segmentation masks as trimaps. In trimap, absolute background is shown as white, absolute foreground is shown as black, and mixed region is shown as gray (see Fig. 11).

Figure 11 Example images from Oxford IIIT Pet dataset.

(A) Abyssinian. (B) Abyssinian trimap. (C) Yorkshire terrier. (D) Yorkshire terrier trimap.

Qualitative results

To demonstrate our data augmentation results qualitatively, we first downloaded sample images from Pxfuel, which provides high-quality royalty-free stock photos. Note that augmented images’ brightness is slightly increased to emphasize the difference between original images and augmented images.

In Fig. 12, augmentation results are shown for man, forest, food, car, and urban images. In the first row, in the augmented male image, the male’s skin color becomes lighter, and the eye color shifts to green. Also, there are some color changes in the background and t-shirt of the man. In the second row, the augmented forest image contains red trees, although there are no red trees in the original image. Here, red trees occur in the augmented image because the original image’s color distribution contains colors towards the red tones in the distribution’s tail. In the third row, in the augmented image, each olive type’s colors in the original image are changed differently while the background is not changed. In the fourth row, the old car’s rust tones in the original image are changed naturally in the augmented image. In the fifth row, the trees and the building roof’s colors become greener with slight color changes in the buildings and the road in the augmented urban image.

Figure 12 Plausible image augmentations.

DPDA results shown in Fig. 12 are all plausible image augmentations. In all these visible results, some image pixel colors are shifted to the tail of the image’s color data distribution. Thus, the image is transformed into a less occurring version of itself. This is quite useful to increase data variability of the training dataset since the proposed data augmentation approach generates fewer occurring images, and thus original dataset is enriched. Therefore, the over-fitting problem is reduced while increasing the training accuracy. Since the image color data guides data augmentation, the algorithm does not require different parameter selections for different images, i.e., images with different content, resolution, or camera characteristics. Accordingly, default DPDA parameters are used for the data augmentations in Fig. 12 (as qualitative experiments) and also for all the quantitative experiments.

Quantitative results

Training a DL network from scratch requires a considerable amount of data and computational power. Therefore, researchers and practitioners with limited data and computational resources prefer to reuse existing DL architectures, which are trained with millions of data and using server farms. This reuse methodology employs a transfer learning approach where a well-proven DL model is fine-tuned with a new dataset (Shao, Zhu & Li, 2015). Pre-training a DL network with transfer learning yields successful results, even with a small train dataset. However, transfer learning provides excellent results if the data and pre-trained model are on a similar domain (Yosinski et al., 2014). A model pre-trained with the Imagenet dataset gives better outcomes for the datasets in the same domain, such as CIFAR-10 or Caltech-101. On the other hand, if the model is tuned using a small amount of training data that is not in a similar domain, the performance benefits of transferring features decrease. So, data augmentation helps increase dataset size and variety to remedy such problems (Shao, Zhu & Li, 2015).

Note that our aim is not to give an extensive study of the architecture of CNNs as done by Szegedy et al. (2015), or He et al. (2016) but to briefly use them for evaluating the performance of the proposed DPDA method in transfer learning settings. Resnet50 (He et al., 2016) and DenseNet201 (Huang, Liu & Weinberger, 2016) network weights trained on the ImageNet are used as starting weights in the classification task since they are widely used in the current studies (Khan et al., 2020). Then the models are fine-tuned during training (Vrbančič & Podgorelec, 2020) since initial layers of CNNs preserve more abstract, generic features. We just copy the weights in convolutional layers rather than the entire network, excluding fully connected layers. MobileNetV2 (Sandler et al., 2018) network weights trained on the ImageNet are used as starting weights in segmentation task as a base model and trained with CNN architecture based on U-Net (Silburt et al., 2019). For all the experiments, we used an Stochastic Gradient Descent (SGD) solver with a momentum of 0.9. Weights are initialized from a Gaussian distribution N(μ,σ) for μ = 0 and σ = 10−2. We found 20 epochs and a batch size of 32 typically sufficient for convergence.

The following methodology was utilized to create train and validation sets for all datasets used in this study. First, we randomly selected 20 images from each class as a validation set and used the same validation set in all tests. Then, we created different train sets in various sizes (N = 20, 30, 40, 50, 60, 70, 80) to investigate the effect of training dataset size on classification performance using the data augmentation approaches. For segmentation tests, we only used the train set size of 80. To avoid sample imbalance in the training datasets, we randomly selected the training datasets in equal numbers from each class. We evaluated the final classification performance of each dataset with the average accuracy over 10 runs. We increased the original training dataset size 5-fold, utilizing random erase (RE), flip image (FI), gamma correction (GC), histogram equalization combined with gamma correction (HE+GC), the proposed DPDA method, and the DPDA method combined with the flip image (DPDA+FI) separately. We implemented color-based augmentation methods as done in CLoDSA (Casado-Garca et al., 2019) library.

Performance analysis

This section presents a performance comparison study using transfer learning with three different DL architectures, namely DenseNet, ResNet, and MobileNetV2. These architectures are trained using transfer learning on original and augmented versions of three datasets. In the experiments, DPDA, DPDA+FI, FI, RE, GC, HE+GC methods are used for the augmentation of images. Baseline performances are obtained by training on the original datasets using the transfer learning approach. Augmentation performances for classification on UC Merced Land-use, Intel Image Classification, and Oxford-IIIT Pet datasets and for segmentation on Oxford-IIIT Pet dataset compared to the baseline performances are presented. For Tables 1–7, the the performance values in bold represent the best performance in its row.

UC merced land-use dataset

First, we compare the performance increase obtained with various data augmentation methods and DPDA on the UC Merced Land-use dataset using DenseNet201 architecture. As seen in Table 1, average accuracy improvement ranges from 1.51% to 4.43%. All data augmentation methods provide performance increase compared to baseline performance for all the train set sizes. However, the DPDA method and the DPDA combined with the flip image consistently provide the best performances in every test. The results also show that data augmentation in data sets with fewer elements contributes more to accuracy. For example, the highest accuracy increase is 6.98% in the training set consisting of 20 images per class, which is obtained with DPDA+FI augmentation.

Table 1 Data augmentation accuracy comparisons (%) in different sizes of datasets (N) using DenseNet201 on UC Merced Land-use dataset.

N	Baseline	DPDA+FI	DPDA	RE	FI	HE+GC	GC	
20	76.35	83.33	82.86	80.23	80.74	79.52	78.96	
30	82.46	88.25	88.09	86.00	85.55	84.52	84.76	
40	84.12	88.33	88.25	86.19	86.11	86.51	85.56	
50	86.27	90.16	90.00	88.41	88.57	87.62	87.93	
60	87.61	91.34	91.43	89.92	89.60	89.76	88.65	
70	89.28	92.62	92.54	91.19	90.71	90.24	90.16	
80	89.60	92.70	92.54	90.95	90.72	90.16	90.24	
Average	85.10	89.53	89.39	87.56	87.43	86.90	86.61	
Increase		4.43	4.29	2.46	2.33	1.81	1.51	
Note:

Performance values in bold represent the best performance in its row.

Next, we compared the performance increase with various data augmentation methods, including the DPDA, using ResNet50 architecture. As seen in Table 2, average accuracy improvement ranges from 2.35% to 5.84%. All data augmentation methods provide performance increase compared to baseline performance for all the train set sizes. However, the DPDA method itself and in combination with the flip image method, consistently provide the best performances in every single test. The results also show that data augmentation in data sets with fewer elements contributes more to accuracy. For example, the highest accuracy increase is 8.49% in the training set consisting of 20 images per class, which is obtained with DPDA+FI augmentation.

Table 2 Data augmentation accuracy comparisons (%) in different sizes of datasets (N) using ResNet50 on UC Merced Land-use Dataset.

N	Baseline	DPDA+FI	DPDA	RE	FI	HE+GC	GC	
20	74.76	83.25	82.86	80.87	80.71	79.84	79.52	
30	80.07	86.67	86.11	83.33	82.78	82.38	81.90	
40	82.62	89.05	88.99	85.95	85.55	84.68	84.12	
50	83.81	88.97	88.29	86.32	86.19	86.97	85.95	
60	85.00	90.48	90.32	87.62	87.93	87.85	87.69	
70	86.66	91.27	91.19	89.87	89.46	88.96	88.57	
80	87.62	91.74	91.67	90.47	90.31	90.18	89.21	
Average	82.93	88.78	88.49	86.35	86.13	85.84	85.28	
Increase		5.84	5.56	3.41	3.20	2.90	2.35	
Note:

Performance values in bold represent the best performance in its row.

Intel image classification dataset

Like the UC Merced Land-use dataset, first, we compare the performance increase obtained with various data augmentation methods, including DPDA, on the Intel Image Classification dataset using DenseNet201 architecture. As seen in Table 3, average accuracy improvement ranges from 2.25% to 4.94%. All data augmentation methods provide performance increase compared to baseline performance for all the train set sizes. However, the DPDA method provides the best performances in every single test. The results also reveal that data augmentation in data sets with fewer elements contributes more to accuracy. For instance, the highest accuracy increase is 7.23% in the training set consisting of 30 images per class, which is obtained with DPDA augmentation.

Table 3 Data augmentation accuracy comparisons (%) in different sizes of datasets (N) using DenseNet201 on Intel Image Classification Dataset.

N	Baseline	DPDA	RE	DPDA+FI	GC	HE+GC	FI	
20	82.00	88.89	87.83	89.17	87.00	87.50	86.33	
30	83.33	90.56	89.16	89.34	89.50	88.33	88.00	
40	86.50	90.00	88.50	89.50	88.33	88.61	86.83	
50	86.66	91.39	90.16	89.67	90.00	89.16	90.33	
60	88.69	92.50	91.83	90.00	90.83	90.83	90.16	
70	88.92	93.33	91.50	90.83	90.83	91.00	90.17	
80	90.16	94.16	92.50	92.50	90.50	90.50	90.17	
Average	86.61	91.55	90.21	90.14	89.57	89.42	88.86	
Increase		4.94	3.60	3.54	2.96	2.81	2.25	
Note:

Performance values in bold represent the best performance in its row.

Next, we compare the performance increase with various data augmentation methods, including the DPDA, using ResNet50 architecture. As seen in Table 4, average accuracy improvement ranges from 1.82% to 4.80%. Every data augmentation methods provide a performance increase compared to baseline performance for all the train set sizes. However, the DPDA method provides the best performances in every single test. The results also indicate that data augmentation in data sets with fewer elements contributes more to accuracy. For instance, the highest accuracy increase is 7.27% in the training set consisting of 20 images per class, which is obtained with DPDA augmentation. This result is also in compliance with the DenseNet comparison study.

Table 4 Data augmentation accuracy comparisons (%) in different sizes of datasets (N) using ResNet50 on Intel Image Classification Dataset.

N	Baseline	DPDA	RE	DPDA+FI	FI	GC	HE+GC	
20	79.67	86.94	85.33	86.57	83.83	84.22	84.61	
30	82.50	88.96	87.50	86.67	86.16	86.44	85.83	
40	84.67	90.28	88.67	89.17	88.66	87.50	86.66	
50	86.83	90.83	89.67	89.17	88.66	88.33	88.22	
60	88.16	91.39	90.00	89.87	89.67	88.33	89.16	
70	88.94	92.78	92.00	90.00	90.83	90.00	89.33	
80	89.33	92.50	92.00	90.83	90.33	88.67	89.00	
Average	85.73	90.53	89.31	88.90	88.31	87.64	87.54	
Increase		4.80	3.58	3.17	2.58	1.91	1.82	
Note:

Performance values in bold represent the best performance in its row.

Oxford-IIIT pet dataset

First, we compare the performance increase with various data augmentation methods, including DPDA, on the Oxford-IIIT Pet dataset using DenseNet201 architecture. As seen in Table 5, average accuracy improvement ranges from 2.34% to 3.34%. All data augmentation methods provide performance increase compared to baseline performance for all the train set sizes while DPDA being superior. The results also reveal that data augmentation in data sets with fewer elements contributes more to accuracy. For example, the highest accuracy increase is 6.68% in the training set consisting of 20 images per class, which is obtained with DPDA+FI augmentation.

Table 5 Data augmentation accuracy comparisons (%) in different sizes of datasets (N) using DenseNet201 on Oxford-IIIT Pet Dataset.

N	Baseline	DPDA	DPDA+FI	HE+GC	FI	GC	RE	
20	81.67	87.89	88.35	86.91	87.43	88.27	86.32	
30	85.26	90.95	89.05	89.91	89.21	88.78	88.13	
40	87.65	90.14	90.27	89.67	89.62	88.83	89.10	
50	88.36	91.13	90.94	90.54	90.67	90.67	90.62	
60	89.85	92.24	92.27	91.21	91.62	91.54	91.81	
70	90.73	92.43	92.51	92.19	92.12	92.51	92.29	
80	90.79	93.10	94.02	93.27	92.91	92.83	92.56	
Average	87.78	91.13	91.06	90.53	90.51	90.49	90.12	
Increase		3.34	3.27	2.75	2.73	2.71	2.34	
Note:

Performance values in bold represent the best performance in its row.

Next, we compare the performance increase with various data augmentation methods, including the DPDA, using ResNet50 architecture. As seen in Table 6, average accuracy improvement ranges from 3.03% to 6.33%. Every data augmentation methods provide a performance increase compared to baseline performance for all the train set sizes. However, the DPDA+FI method provides the best performances in every single test. The results also show that data augmentation in data sets with fewer elements contributes more to accuracy. For instance, the highest accuracy increase is 12.66% in the training set consisting of 20 images per class, which is again obtained with DPDA+FI augmentation.

Table 6 Data augmentation accuracy comparisons (%) in different sizes of datasets (N) using ResNet50 on Oxford-IIIT Pet Dataset.

N	Baseline	DPDA+FI	DPDA	RE	FI	GC	HE+GC	
20	67.74	80.40	79.05	77.08	78.27	72.94	72.51	
30	75.46	82.83	82.48	82.27	81.08	81.40	79.78	
40	78.51	85.54	84.70	82.75	82.70	82.91	82.99	
50	80.72	86.62	86.48	84.64	85.67	84.29	84.16	
60	84.01	86.70	88.73	87.54	85.81	86.75	85.43	
70	84.91	90.08	90.00	89.94	88.24	88.40	86.57	
80	85.85	89.32	89.34	88.83	89.00	88.51	87.00	
Average	79.60	85.93	85.83	84.72	84.40	83.60	82.63	
Increase		6.33	6.23	5.12	4.80	4.00	3.03	
Note:

Performance values in bold represent the best performance in its row.

We used the U-Net architecture on top of the MobileNetV2 architecture for the Oxford-IIIT Pet Dataset segmentation experiments. DPDA provides by far the best performance in these experiments (see Table 7). The accuracy improvement obtained with DPDA is 8.49%. The second highest accuracy improvement achieved with the FI augmentation is 0.81%, which is much less than the DPDA accuracy improvement. On the other hand, every data augmentation method does not provide a performance increase compared to baseline performance. For instance, the GC decreases the accuracy by −2.20%.

Table 7 Segmentation performance comparisons using MobileNetV2+U-Net on Oxford-IIIT Pet Dataset.

	Baseline	DPDA	DPDA+FI	FI	RE	HE+GC	GC	
Average Accuracy	80.82	89.31	82.48	81.64	80.59	80.44	78.63	
Accuracy Increase		8.49	1.66	0.81	0.13	−0.38	−2.20	
Note:

Performance values in bold represent the best performance in its row.

The above results indicate that models trained with the augmented UC Merced Land-use, Intel Image Classification, Oxford-IIIT Pet datasets, with the DPDA method, significantly improve classification performance. Besides, DPDA also provides superior performance in an image segmentation task. Thus, we can infer that the proposed DPDA method can improve DL performance for different datasets, different DL architectures (ResNet, DenseNet, and MobileNetV2), and different image analysis tasks.

Execution time analysis

There are n pixels in an image. During the execution of DPDA, for each pixel, we find a path with a length of up to L/2. For each point in a path, the nearest neighbor search using FLANN method (Muja & Lowe, 2014) is done to retrieve k neighbor points. Our data is 3 channel so dimension d is 3. For each image, the FLANN tree is constructed for once where tree construction has a computational complexity of O(ndKI(logn/logK)), where I is a maximum number of iterations, and K is the branching factor. We used exact search in FLANN, which leads to O(Md(logn/logK)) for single nearest neighbor search where M is a maximum number of points to examine. However, we need to do a separate neighbor search for L/2 times for n pixels, which leads to nL/2 neighbor search operations. Thus, computational complexity of the all neighbour search operations is O(nLMd(logn/logK)). Therefore, computational complexity of the DPDA is O(nd(KI+LM)(logn/logK)) including tree construction and neighbor search operations.

The average execution time for 10 augmentations of DPDA, RE, GC, and FI methods concerning image size (# of pixels) is shown in Fig. 13. Although FLANN provides efficient nearest neighbor search operations, as shown in Fig. 13, the execution times of the DPDA method are longer compared to RE, GC, and FI methods. Fortunately, in DL training, images are generally in small sizes, i.e. 435 × 387 for Oxford dataset, 250 × 250 for UCMerced dataset, and 150 × 150 for Intel dataset (see Fig. 13).

Figure 13 Execution time (in log-scale) with respect to image size (# of pixels, n).

Discussion & Future Works

The proposed DPDA method employs a distribution preserving approach to create plausible variants of a given image, as shown in qualitative and quantitative results. These augmented images enrich the training dataset so that the over-fitting problem is reduced while higher training accuracies are obtained. Obtained augmentation performance is demonstrated on UC Merced Land-use, Intel Image Classification, and Oxford-IIIT Pet datasets for classification and segmentation tasks. These experiments show the superiority of the proposed DPDA method compared to commonly used data augmentation methods such as image flipping, histogram equalization, gamma correction, and random erasing. We also combined our DPDA method with a geometric data augmentation method (flip), and in most cases, the performance of DPDA is slightly increased. This shows that the DPDA method can be combined with other data augmentation methods to increase performance further. Therefore, it is evident that DPDA is a good candidate for data augmentation tasks in different scenarios. This is consistent with the research outcomes in the literature where various data augmentation methods provide performance improvement in numerous machine learning tasks and datasets (Shorten & Khoshgoftaar (2019)). Although the proposed method provides outstanding data augmentation capabilities, there is still room for further improvements. These improvements can be divided into three groups: computational efficiency improvements, augmentation performance improvements (reflecting on DL training), and usage dissemination improvements.

Data augmentation methods are generally fast, while the DPDA method is not as fast as its competitors. The main reason for this speed bottleneck is the computational burden of neighbor search, which is also the reason for the slowness of mean-shift-based clustering or filtering methods. This bottleneck can be alleviated by changing FLANN with a faster or a specifically designed neighbor search method. Additional speed-ups can be obtained using CPU and GPU parallelization techniques since an image contains lots of pixels, and finding density decreasing path for each pixel is independent of other pixels that can be done in parallel. Since using GPU is a common approach for DL training, GPU parallelized DPDA method will not cause extra hardware procurement on its user.

Performance of the DPDA method can be increased using spatial regularization, i.e., using graph-cut, dealing with blocking artifacts due to JPEG compression. Similar images can be retrieved, and their color data can be added to the image color data to be augmented, which may increase the quality and variety of the color data distribution especially if the image size is small. DPDA uses Perlin noise to create different augmentations from a single density decreasing path per pixel. However, Perlin noise is spatially smooth approach but still a purely random one. Instead, an image can be segmented into background and foreground objects then randomization can be done in an object-wise manner.

DPDA code can be extended to multispectral and hyperspectral images, which have 4 or more channels. Additionally, DPDA is not limited to the augmentation of images and can be easily adapted to augment any training data since it already works in a feature space. This is quite useful for training traditional machine learning methods that generally work on data with already extracted features. Furthermore, DPDA can be ported to Python for easy integration with current Python-based DL frameworks.

As a future study, in addition to various performance improvements and support for augmentation in feature space, we plan to improve computational efficiency using special techniques and data structures with a parallelized implementation in Python.

Conclusions

In this paper, a novel distribution-preserving data augmentation (DPDA) method that creates plausible variations of the given image is presented. There is no study using a distribution-preserving approach that creates plausible image variations to the best of our knowledge. The proposed method employs density decreasing direction to create paths from colors of the original pixels to the tails of the image data distribution. We achieved this by regularizing the opposite of the mean-shift direction with length and orientation constraints. Finally, we developed efficient mechanisms to obtain these density decreasing paths, fused with Perlin noise results to create as many augmented images as desired.

The proposed method’s performance is presented in a transfer learning scenario using three different DL architectures: DenseNet, ResNet, and MobileNetV2. These DL architectures are trained with millions of color images, where we used transfer learning to adapt these models to different problem domains. We tested the DPDA for classification on the UC Merced Land-use, Intel Image Classification, and Oxford-IIIT Pet datasets and image segmentation on the Oxford-IIIT Pet dataset. Note that, DenseNet, ResNet, and MobileNetV2 are trained with side-view commodity camera images, namely ImageNet. On the other hand, the UC Merced land-use dataset is obtained from nadir as over-head imagery (that can be acquired using airborne and spaceborne platforms). Also, the resolution and camera characteristics of the ImageNet dataset are pretty different from the resolution and camera characteristics of the UC Merced Land-use dataset. Nevertheless, transfer learning able to cope with this challenging adaptation. However, the UC Merced land-use dataset’s size is small, limiting the applied transfer learning schema’s adaptation performance. This is a common scenario since companies or institutions develop pre-trained models with large datasets and substantial computational resources. Despite this, researchers who use these pre-trained models with transfer learning to adapt them to their problem domain generally have small datasets and scarce computational resources. In this study, the transfer learning performance is further increased using data augmentation methods such as the proposed DPDA, image flipping, histogram equalization, gamma correction, and random erasing. On the other hand, for image classification and segmentation tasks, the proposed DPDA method consistently shows superior performance compared to commonly used data augmentation methods on different datasets and different training sizes using three different DL architectures. Therefore, we concluded that the proposed DPDA method provides successful data augmentation performance.

Although the proposed method provides superior data augmentation capabilities, there is still room for further improvements. However, we did not implement these improvements since we want to present our novel density-preserving data augmentation idea’s baseline performance in its simplest form. Nevertheless, possible improvements and future studies are shared in the ‘Discussion & Future Works’ section. Among these possible future studies, improving the computational efficiency of the proposed DPDA is the most important one since high computational complexity seems to be the most significant disadvantage of the proposed method. As a final remark, although we presented our DPDA method as an image augmentation study, it is not limited to images and can work for all kinds of the dataset with already extracted features since it works in feature space. This is an excellent property of the proposed DPDA method since most image data augmentation methods are only limited to the image domain.

Supplemental Information

Supplemental Information 1 Data augmentation accuracy comparisons using DenseNet201 on UC Merced Land-use dataset (for Table 1).

Click here for additional data file.

Supplemental Information 2 Data augmentation accuracy comparisons using ResNet50 on UC Merced Land-use Dataset (for Table 2).

Click here for additional data file.

Supplemental Information 3 Data augmentation accuracy comparisons using DenseNet201 on Intel Image Classification Dataset (for Table 3).

Click here for additional data file.

Supplemental Information 4 Data augmentation accuracy comparisons using ResNet50 on Intel Image Classification Dataset (for Table 4).

Click here for additional data file.

Supplemental Information 5 Data augmentation accuracy comparisons using DenseNet201 on Oxford-IIIT Pet Dataset (for Table 5).

Click here for additional data file.

Supplemental Information 6 Data augmentation accuracy comparisons using ResNet50 on Oxford-IIIT Pet Dataset (for Table 6).

Click here for additional data file.

Additional Information and Declarations

Competing Interests

Author Contributions

Data Availability

1 Pxfuel: https://www.pxfuel.com/ (Royalty-free stock photos free & unlimited download).

The authors declare that they have no competing interests.

Nurdan Ayse Saran conceived and designed the experiments, analyzed the data, performed the computation work, authored or reviewed drafts of the paper, and approved the final draft.

Murat Saran conceived and designed the experiments, performed the experiments, analyzed the data, performed the computation work, prepared figures and/or tables, authored or reviewed drafts of the paper, and approved the final draft.

Fatih Nar conceived and designed the experiments, analyzed the data, performed the computation work, prepared figures and/or tables, authored or reviewed drafts of the paper, and approved the final draft.

The following information was supplied regarding data availability:

The Windows (Visual Studio) project repository is available at GitHub:

https://github.com/msaran1923/dpda.

The Linux project repository is available at GitHub:

https://github.com/msaran1923/dpda-linux.

The UC Merced Land Use Dataset is available at: http://weegee.vision.ucmerced.edu/datasets/landuse.html.

The Intel Image Classification is available at Kaggle: https://www.kaggle.com/puneet6060/intel-image-classification.

The Oxford-IIIT Pet Dataset is available at: https://www.robots.ox.ac.uk/~vgg/data/pets/.

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
