# Peer review of "Distribution-preserving data augmentation"

_PeerJ Computer Science, doi:10.7717/peerj-cs.571_

## Round 0.1 · original submission · Major Revisions

Please consider including other data augmentation methods in the comparison.

Reviewer 1 ·

Basic reporting

The paper presents a color-based data augmentation method based on distribution-preserving color that creates plausible image variation by shifting pixel colors to another point in the image distribution. Although, this research may be interesting, in my opinion the paper has several drawbacks that should be addressed.

The introduction is lengthy and does not focus on the contributions of the paper. Moreover, the paper is sometimes repetitive. For instance, the idea of using the mean-shift process by Fukunaga and Hosteler and Comaniciu and Meer is mentioned in Line 108, in Line 134, in Line 169, and in Line 199. Another example is the "experimental setup" section which repeats ideas about the usefulness of transfer learning instead of directly explaining the “experimental setup”. In addition, the experimental setup is repeated in the Test Settings and Quantitative Results sections.

The sentence “the idea also fits in the feature domain, so it can be applied to the features directly”, in Line 119, should be removed from the introduction and, if considered necessary, explained in more detail in the material and methods sections.

Experimental design

My main concern is that the method is compared with a unique augmentation method (random erasing) on a unique dataset (UC Merced Land-use). But, on the one hand, the authors do not explain why this dataset is chosen and why this dataset can benefit for the proposed method. On the other hand, the authors do not explain why the random erasing augmentation method is used to be compared with the proposed method. Random erasing is a method in which training images are generated with various levels of occlusion, which reduces the risk of over-fitting and makes the model robust to occlusion. The goals of both methods seem to be very different.

In my opinion, the authors should try to strive to find some dataset in which the proposed augmentation method could be essential. Furthermore, the results of the proposed method should be compared with more similar methods based on color transformations.
That is proposed in the abstract (Line 25): “…superior performance of existing color-based data augmentation methods”. Libraries such as albumentations or CLoDSA implements several color based transformations which can be used in such as comparison.

The authors propose an image pyramid to increase the number of extracted features. Then, two figures are included showing the effect of this increase in features. The authors explain (lines 166-167) that “the effect of image quantization errors are reduced, which demonstrates the benefit of the image pyramid approach”. Can the authors explain to what extent these errors are reduced?

Figure 11 shows that the execution time of the method is very large for large images. A comparison of execution times of the proposed method and other color-based data augmentation methods used in the Performance Analysis section should be included.

Validity of the findings

There is not an actual Discussion section in which a comparison of the DPDA method and other similar methods proposed in the literature. The first paragraph of this section only includes a repetition of some characteristics of the DPDA method. Furthermore, it is suggested that “the method obtain superior performance compared to other data augmentation approaches, due to plausible augmentation”. But, as I previously mentioned, the method is only tested on a unique dataset and compared with a unique data augmentation method. Moreover, it is not explained why this dataset can benefit of such as plausible augmentation. Finally, the rest of the Discussion section propose future work. In my opinion, some of this future work could try to be addressed before the method is published in a journal. In particular, methods for speeding up the proposal.

Additional comments

Minor points:
- Line 171 change pdf by PDF
- Last sentence in line 221 -> moidified
- The link to the developed library should be included in the paper.

·

Basic reporting

The paper is well-written and the explanations are clear. In general, the paper provides enough references. However, I think that in the introduction (paragraph starting in Line 56) some recent data augmentation methods such as MixUp (Zhang et al, 2017), CutMix (Yun et al. 2019), CutOut (De Vries and Taylor. 2017) or AugMix (Hendrycks et al. 2019) should also be included.

References.
Zhang et al. 2017. mixup: Beyond empirical risk minimization. https://arxiv.org/pdf/1710.09412.pdf.
De Vries and Taylor. 2017. Improved Regularization of Convolutional Neural Networks with Cutout. https://arxiv.org/pdf/1708.04552v2.pdf
Yun et al. 2019. CutMix: Regularization Strategy to Train Strong Classifiers with Localizable Features. https://arxiv.org/abs/1905.04899
Hendrycks et al. 2019. AugMix: A Simple Data Processing Method to Improve Robustness and Uncertainty. https://arxiv.org/abs/1912.02781

Experimental design

My main issue with this paper is that the experiments are quite narrow. The authors have focused on just one dataset, so it is not possible to know if this data augmentation regime serves to several contexts. It would be interesting to know if the data-augmentation procedure presented in the paper works not only for image classification but also for other computer vision problems such as object detection and instance segmentation.

Related to the previous point, the authors only compare their approach with the random erase augmentation method since they claimed that this is the state-of-the-art. But, this augmentation method is not the state-of-the-art for all datasets. So, the authors should clarify this. Moreover, it would be necessary to compare the results with other augmentation techniques, specially with color transformations since the method presented in the paper could be framed in this kind of augmentation. Finally, the authors claim in Line 388 that their method can be easily combined with other color or geometric augmentations to increase the performance of models, but this is not proved at all.

Validity of the findings

The results achieved by the authors are not conclusive since they have only focused on a particular dataset and task. Moreover, a thorough comparison with other methods is necessary.

Additional comments

A plus of this work is that they provide the code for their work. However, there is not any instruction about how to install their program, or to reproduce the experiments of the authors.
An additional issue is that the code is implemented in C++ and it seems to require Visual Studio. This has two problems. First, most people working nowadays in deep learning use Python as programming language, so it is not possible to directly use your code (this is admitted by the authors, but it is important to work in this direction). Similarly, I think that it is not possible to install Visual Studio in Linux so many users will not be able to use it. At least the authors should indicate the requirements to run your code. Moreover, it will be helpful if the authors provide the link to the code in the manuscript. Finally, it would be interesting to integrate your methods in libraries such as Albumentations (https://albumentations.readthedocs.io/) or Clodsa (https://github.com/joheras/clodsa).

---

## Round 0.2 · Minor Revisions

In general, reviewers agree this new version of the paper has many improvements and they have only minor suggestions to accept it. Please take them into account in the next version of the paper

·

Basic reporting

The issues reported in the previous version have been addressed.

Experimental design

A minor issue with the Intel Image Classification dataset is that the authors did not include the results obtained using DPDA+FI. The same happens in Table 7. Please be consistent when presenting the results from Tables 1 to Table 7.

Validity of the findings

The approach presented by the authors seems to consistently improve the resulsts in several datasets.

Additional comments

The authors have addressed all my concerns, and it is great that they provide the code for both Windows and Linux. As the authors suggest in the response porting their code to Python will be quite time consuming and is independent from the presented method.

Reviewer 3 ·

Basic reporting

- The authors have improved the manuscript in overall following the suggestions made by the reviewers.

- Although source code and data source links were provided, authors should consider Python to reach a bigger audience.

Experimental design

- The experimental study was extended by including 2 datasets and a segmentation analysis, which was also suggested in the previous revision.

- There is a concern in Figure 13. The authors should have plotted the execution time regarding the other methods. According authors, there is a great gap between compared methods and the proposal. A section where a balance is made between performance and runtime metrics is needed, e.g. authors could elaborate a new metric where both mentioned metrics are merged.

Validity of the findings

- The authors provided an extended discussion about the proposal and the possibilities to combined it with other data augmentation methods.

- Finally, the authors should consider the execution time as an important issue. So far, the time reported in section “Execution Time Analysis” was for 1 image. An analysis about how long a complete execution takes in every dataset should be included.

---

## Round 0.3 · accepted · Accept

After reading your response to the reviewers and looking at this new version, I think the document is ready for publication. Congratulations.